# Synthesis and Characterization of Cellulose Diacetate-Graft-Polylactide via Solvent-Free Melt Ring-Opening Graft Copolymerization

**DOI:** 10.3390/polym15010143

**Published:** 2022-12-28

**Authors:** Shiyou Zhao, Jin Li, Lifeng Wu, Ming Hua, Changmei Jiang, Ying Pan, Lirong Yao, Sijun Xu, Jianlong Ge, Gangwei Pan

**Affiliations:** 1National & Local Joint Engineering Research Center of Technical Fiber Composites for Safety and Protection, School of Textile and Clothing, Nantong University, Nantong 226019, China; 2CSIC Pacli (Nanjing) Technology Co., Ltd., Nanjing 211106, China

**Keywords:** cellulose diacetate, grafting rate, process condition, polylactide, thermal processing temperature

## Abstract

Cellulose diacetate (CDA) and L-lactide (L-LA) were used to prepare CDA−g−PLLA with a low glass transition temperature under different process conditions. Given the high glass transition temperature (T_g_) of CDA, the thermal processing performance of CDA is poor, which greatly limits its application fields. To decrease the T_g_ of CDA, graft copolymerization was used in this research. A CDA−g−PLLA graft copolymer was synthesized by grafting CDA with L-LA under different reaction conditions using stannous octanoate as the catalyst and variations in the grafting rate under different reaction conditions were compared. The chemical structure and crystal structure of the CDA−g−PLLA were investigated, and thermal properties were also studied. The results showed that the grafting rate was the highest at the L-LA/CDA mass ratio of 4:1 under a reaction temperature of 150 °C for 90 min, and no poly-L-lactide (PLLA) homopolymer was found among the CDA−g−PLLA graft copolymers after purification. The T_g_ of CDA−g−PLLA was 54.2 °C, and the initial temperature of weightlessness of CDA−g−PLLA was 218.7 °C. The regularity of the original CDA molecular chains was destroyed after grafting PLLA molecular chains. In this research, we investigated the optimal grafting conditions for CDA−g−PLLA and the CDA−g−PLLA had a low T_g_, which improves the thermal processing performance of CDA and broadens its application prospects in the industry.

## 1. Introduction

In recent years, environmentally friendly renewable materials have been increasingly drawing people’s attention [1,2,3,4]. Cellulose diacetate (CDA) is a commercially important biodegradable acetylated cellulose with an average degree of substitution (DS) of 2.2–2.7. As a cellulose derivative, CDA is already being processed into fibers, films, and coatings due to its excellent biocompatibility and biodegradability [5,6,7,8]. In addition, CDA is thermoplastic with the addition of plasticizers and can be prepared into various plastic forms by thermoplastic processing. Given its good transparency, selective permeability, and high rigidity, CDA has been widely used in various fields [9,10,11].

Given the strong intermolecular interactions in CDA, the molecular chains of CDA are difficult to move, thus resulting in a high glass transition temperature (T_g_). However, the high T_g_ of CDA leads to poor thermal processability. In addition, CDA products usually have low mechanical properties. In terms of poor thermoplasticity and low strength, researchers have undertaken substantial research on the blending modification of CDA [12,13,14]. Bao et al. [15] proposed the preparation of CDA films via a solvent casting method with diethyl phthalate as the plasticizer, which could decrease the T_g_ of CDA. Vu Thanh et al. [16] blended ethyl diacetate and ethyl triacetate as plasticizers with CDA and obtained products with high toughness. Kenaf fiber sized with poly(vinyl alcohol) was used to reinforce CDA, resulting in improved mechanical and thermal properties of CDA. Moreover, the tensile strength and elastic modulus of the composites increased by nearly two and three times, respectively, when the kenaf fiber content was 30 wt% [17]. Although the blending modification could improve the material’s properties, some problems, including poor compatibility, plasticizer precipitation, and degradation, still need urgent attention [18,19,20].

The modification of CDA by grafting can overcome the shortcomings of traditional plasticizers while introducing the properties of grafting groups, thereby providing an idea on how to improve the thermoplastic and mechanical properties of CDA simultaneously [21]. Iji et al. [22] described the bonding of CDA with a cardanol group that had a terminal long alkyl chain, and by modifying the DS of cardanol, cardanol-bonded CDA with T_g_ as low as 140 °C could be produced. Liang et al. [23] grafted poly(ethylene terephthalate) (PET) onto the CDA backbone and prepared CDA-g-PET with superior mechanical and thermal processing properties compared with those of CDA. The introduction of biopolymer groups on the CDA molecule decreases the CDA processing temperature and imparts certain properties to the graft. Thus, the selection of graft is also important.

In the field of thermoplastics, poly(lactic acid) (PLA) is known for its biodegradability and biobased properties. PLA has two isotactic stereoisomers due to lactic acid’s chirality molecule, poly(L-lactic acid) (PLLA) and poly(D-lactic acid) (PDLA). Under certain conditions, stereocomplex (sc) crystals form between PLLA and PDLA molecular chains, and the resultant stereocomplex poly (lactic acid) (scPLA) can endow PLA-based materials with strong mechanical characteristics and heat resistance. [24,25,26]. It is a green thermoplastic polyester that combines biocompatibility with bioresorbability. It is widely used in medical, pharmacy, and agricultural fields, and it is currently considered the most prospective biodegradable polymer [27,28,29,30,31]. Xu et al. [32] synthesized a series of CDA-grafted PLA (CDA−g−PLA) copolymers with various side chain lengths, and it was found that the T_g_ of the CDA-g-PLA copolymers was highly dependent on the degree of polymerization of the PLA molecule chain, and the CDA−g−PLA copolymers with the degree of polymerization of the grafted PLA of 3–9 could be directly processed into transparent plastics without any external plasticizer. However, the relationship between the grafting rate and grafting conditions remains unclear and must be further investigated.

The ring-opening polymerization of L-lactide (L-LA) with stannous octanoate as the catalyst and CDA as the grafting backbone was used to synthesize a copolymer of CDA−g−PLLA in this study. The effects of the L-LA/CDA mass ratio, reaction temperature, and reaction time on the monomer conversion and grafting rate were investigated. Fourier transform infrared spectroscopy (FTIR) and ^1^H nuclear magnetic resonance spectra (^1^H NMR) were used to explore the chemical structure of the graft copolymers. Thermogravimetric Analysis (TGA) and differential scanning calorimetry (DSC) were used to study the thermal properties of the graft copolymers, and X-ray diffraction (XRD) was also used to analyze the crystal structure of the graft copolymers.

The purpose of this study is to evaluate the thermal processing performance of CDA−g−PLLA graft copolymers, so that CDA can be thermally processed more easily without adding other inorganic small molecule plasticizers. It also provides process reference for the subsequent synthesis of CDA−g−PDLA and CDA−g−scPLA materials.

## 2. Materials and Methods

### 2.1. Experimental Materials

CDA with a DS of 2.46 was obtained from Nantong Acetate Co. L-LA was purchased from Shandong Longjing New Material Technology Co. Stannous octanoate [Sn(Oct)_2_] (98.5% purity) was purchased from Aladdin Biochemical Technology Co. (Shanghai, China). Trichloromethane was purchased from Sinopharm Chemical Reagent Co., Ltd., and anhydrous methanol was purchased from Shanghai Lingfeng Chemical Reagent Co. The toluene was provided by Sinopharm Chemical Reagent Co. (Shanghai, China).

### 2.2. Synthesis and Purification of CDA−g−PLLA Copolymers

#### 2.2.1. Synthesis of CDA−g−PLLA Copolymers

First, CDA was crushed into a powder (the approximate particle size was 359.04 μm) by a crusher, and then the CDA powder and L-LA were dried at 60 °C in a vacuum drying oven. When the temperature in the reactor (LB1000, LABE Instrument Co., Ltd., Shanghai, China) increased to the temperature that was required for the experiment, CDA and L-LA were placed in the reactor according to the feeding mass ratio. The reactants was stirred and dissolved under nitrogen protection for 40 min, and Sn(Oct)2 accounting for 2% of CDA’s quality was added to the reactor. Subsequently, the reaction was continued under nitrogen protection and finished after reaching the reaction time. When the reactor cooled to room temperature, the solid reactants were removed from the reactor.

#### 2.2.2. Purification of CDA−g−PLLA Copolymers

The solid reactants were dissolved in trichloromethane and stirred with a magnetic stirrer for 24 h. After completely dissolving, the solution was slowly poured into anhydrous methanol for precipitation, and the solid products were dried in a vacuum drying oven (DZF-6050, Jinghong Experimental Equipment Co., Shanghai, China) at 60 °C. Solid crude products were finally obtained after filtration. With toluene used as the solvent, the solid crude products were purified in a Soxhlet extractor and refluxed for 24 h. After drying, the CDA−g−PLLA graft copolymers were obtained.

### 2.3. Characterization

#### 2.3.1. Analysis of Grafting Rate

The obtained samples were calculated using the weighing method, and each experiment was repeated three times. Valid data were averaged to investigate the effects of the feeding mass ratio of L-LA/CDA, reaction time, and reaction temperature on the grafting rate and monomer conversion of CDA−g−PLLA.

The grafting rate (G%) and monomer conversion (C%) of the grafting products were calculated by Equations (1) and (2):(1)$$G\%=\frac{W_{3}-W_{0}}{W_{0}}\times 100\%$$
(2)$$C\%=\frac{W_{2}-W_{0}}{W_{1}}\times 100\%$$
where *W*_0_ is the feeding amount of CDA, *W*_1_ is the feeding amount of L-LA, *W*_2_ is the weight of crude product, and *W*_3_ is the weight of pure CDA−g−PLLA graft copolymer.

#### 2.3.2. FTIR

CDA and purified CDA−g−PLLA were scanned by the ATR method at room temperature using an IS50+IN10 FTIR spectrometer from Nicolet, USA for the samples in the range of 500–4000 cm^−1^ with an average of 32 scans.

#### 2.3.3. ^1^H NMR

The obtained samples were measured using an AVANCE III HD 400 MHz NMR spectrometer from Bruker, Switzerland, using deuterated chloroform as solvent and MestReNova software to correct the chemical shifts.

#### 2.3.4. TGA/DTG

Each sample of 5 mg was placed in a ceramic crucible and then tested for thermal stability using a thermogravimetric analyzer model TG209F3 from NETZSCH, Germany, at a temperature of 40–600 °C with a ramp-up rate of 10 °C/min, using nitrogen as a protective gas.

#### 2.3.5. DSC

A differential scanning calorimeter DSC200F3 from Netzsch, Germany was used to measure the graft copolymers. About 5–10 mg of pure graft copolymers were added to the aluminum crucible. Then, the temperature was cooled down from 250 °C to 30 °C at a rate of 20 °C/min and finally heated again to 250 °C to record the second heating curve.

#### 2.3.6. XRD

The graft copolymers were tested using Rigaka’s X-ray diffractometer (Ultima IV, Tokyo, Japan). The testing conditions were as follows: CuKa rays were used as the radiation source, Ni was filtered, the wavelength was 0.1542 nm, a scan voltage of 40 kV and a scanned current of 40 mA were used, and a scan speed of 3°/min was used at a scanning interval of 0.02° and diffraction angle (2θ) of 5–50° for graft copolymers.

## 3. Results and Discussion

### 3.1. Influencing Factors of Graft

#### 3.1.1. Effect of Feeding Mass Ratio of L-LA/CDA

CDA and L-LA were grafted under a nitrogen atmosphere. Sn(Oct)_2_ was used as a catalyst, and the reaction was conducted at 130 °C for 1 h [33]. The effect of feeding mass ratio on the grafting rate and monomer conversion rate of CDA−g−PLLA was studied.

Figure 1 shows that the grafting rate increases and then decreases as the mass ratio of L-LA increases and reaches the maximum at 4:1. According to this dataset, the lowest grafting rate was 133.3%, and the highest grafting rate reached 207.5%. This result was obtained probably because the grafting rate increased as the L-LA increased, and the chance of effective collision between L-LA and CDA increased under the action of stirring. Thus, it was more helpful for the grafting reaction. The grafting rate decreased after the feeding mass ratio was higher than 4:1 due to the excess addition of L-LA. In the event of stirring, the chances of L-LA colliding with another L-LA are much greater than the chances of L-LA colliding with CDA, so the chance of its homopolymerization was greater than that of grafting with CDA, resulting in more PLLA oligomers. Although the amount of L-LA increased and the crude product increased slightly, the increase in crude product was much smaller than the amount of L-LA feeding, and the smaller-molecular-weight PLLA could not be precipitated completely in anhydrous methanol; thus, the monomer conversion rate gradually decreased.

#### 3.1.2. Effect of Reaction Time

L-LA and CDA were reacted at the reaction temperature of 130 °C with a feeding mass ratio of 4:1. The effect of reaction time on the grafting rate and monomer conversion rate of CDA−g−PLLA was studied.

As shown in Figure 2, the grafting rate and monomer conversion rate increased and then decreased with the increase in time and reached the maximum value at a reaction time of 90 min. Figure 2 shows the highest grafting rate reached 235% and the highest monomer conversion rate was 87.5%. This result was obtained probably because the grafting reaction was not completely reacted before 90 min. The reaction was gradually completed as the time increased. The viscosity of the fluid in the reactor gradually increased with the increase in time after 90 min. Thus, the reaction was difficult to continue, and the ester exchange also appeared in the reaction system at a high temperature, resulting in by-products. Raw materials and reaction products underwent a certain degree of thermal degradation with the increase in time. As the rate of thermal degradation was greater than the rate of grafting, the grafting rate and monomer conversion rate gradually decreased.

#### 3.1.3. Effect of Reaction Temperature

L-LA and CDA were reacted for 90 min at a feeding mass ratio of 4:1 to study the effect of reaction temperature on the grafting rate and monomer conversion rate of CDA−g−PLLA.

Figure 3 shows that the grafting rate and monomer conversion rate exhibited a trend of increasing and then decreasing with the increase in reaction temperature, and the maximum value was reached at 150 °C. The grafting rate of graft copolymers decreased from the highest 270.3% to 256% with increasing temperature and the monomer conversion rate increased from 83.5% to 87.2% and then decreased to 86.2%. This is because the activity of the catalyst was gradually enhanced with the increase in reaction temperature. Thus, the reaction could be conducted more completely. Although the rate of thermal degradation and grafting reaction increased with the increase in reaction temperature, the former increased more rapidly than the latter. Moreover, the increase in temperature led to more intense side reactions within or between molecules, which also resulted in a decrease in the grafting rate and monomer conversion. In addition, the color of the graft copolymers gradually changed from light yellow to dark brown and finally to black as the temperature increased.

### 3.2. FTIR Analysis

The –OH stretching vibration peak of CDA at around 3500 cm^−1^ can be seen in Figure 4, whereas the –OH stretching vibration peak at 3500 cm^−1^ tended to level off in the FTIR spectra of the CDA−g−PLLA graft copolymers, which was due to the substitution of –OH on the CDA molecular chain by the newly grafted PLLA molecular chain. In addition, the –C=O vibrational peak at 1740 cm^−1^, the –CH_3_ stretching vibrational peak at 1370 cm^−1^, and the C–O–C stretching vibrational peak at 1210 cm^−1^ were all characteristic peaks of CDA. Compared with the FTIR spectra of CDA, the CDA−g−PLLA graft copolymers showed a blue shift of the –CH_3_ stretching vibration peak at 1430 cm^−1^ [32]. New asymmetric C–O–C stretching vibration peaks can be seen in the ester group at 1091, 1131, and 1186 cm^−1^. A considerably intensive absorption peak of the –CH at 3000 cm^−1^ was found. At the same time, the oscillating vibration peak of –CH_2_ appeared at 737 cm^−1^, which confirmed the formation of side chain of PLLA [34]. Meanwhile, the FTIR spectra of the CDA−g−PLLA graft copolymers also showed the presence of the C=O stretching vibration peak at 1740 cm^−1^ and –CH_3_ stretching vibration peak at 1370 cm^−1^ with remarkable enhancement compared with the characteristic peaks of CDA, and these results can prove that PLLA was successfully grafted to the CDA molecular chain.

### 3.3. ^1^H NMR Analysis

The chemical structure of the graft was further determined using ^1^H NMR in Figure 5, and the –CH (b) was in the PLLA structure unit. The –CH_3_ (c) and –CH (d) in the PLLA structure unit contained hydroxyl end groups at δ = 1.5 and 4.4 ppm and the impurity peak (toluene) at δ =2.4 ppm. Some resonance peaks are apparent for the methyl proton of the acetyl group in CDA at 1.8–2.1 ppm € [35]. Meanwhile, the carboxyl-containing end group peaks of linear PLLA did not appear at δ = 1.3–1.4 ppm and 4.9–5.0 ppm. Thus, no homopolymer was present in the product. Therefore, the characterization of ^1^H NMR can further prove that the L-LA underwent ring-opening polymerization and grafting onto the CDA molecular chain.

Then, estimates of the molar substitution (MS), the average length of the grafted chains (DPs), and the degree of lactyl substitution (lactyl DS) were estimated as previously reported [18]:(3)$$MS=DS_{CDA}\left(A_{A}+A_{C}\right)/A_{e}$$
(4)$$lactyl\;DS=DS_{CDA}A_{C}/A_{e}$$
(5)DPs=MS/(lactyl DS)
where *A* is the area of the corresponding NMR peak, and 2.46 is the degree of acetyl substitution in CDA. Estimates have been based on previous assumptions that the degree of substitution of CDA remained invariable during the reactive process.

As can be seen in Table 1, with the increase of reaction temperature, lactyl DS in the graft copolymers first increased, then decreased, and reached the maximum at 150 °C, indicating that the polymerization reaction was most efficient at this temperature. This may be because the increase in temperature increases the activity of the catalyst and thus the reaction efficiency, but too high temperature will aggravate the thermal degradation and thus reduce the lactyl DS. This result also verified the conclusion obtained in Section 3.1.3.

### 3.4. TGA and DTG Analysis

As shown in Figure 6, the initial thermal decomposition temperature of CDA is 341.7 °C, and the termination temperature of thermal decomposition is 419.6 °C. Moreover, the TGA curve of CDA exhibited only a clear downward trend (corresponding to only one peak in the DTG plot), which indicated that the thermal degradation of CDA was completed in one step. There were two clear downward trends that were observed in the TGA curves of the graft copolymers (corresponding to two peaks in the DTG plots), indicating that the degradation of CDA−g−PLLA was divided into two stages. The first stage was the thermal degradation of the PLLA molecular chain, and the second stage was the thermal degradation of the CDA molecular chain; thus, a certain microscopic phase separation occurred within the graft copolymers.

Comparing the TGA and DTG plots of CDA−g−PLLA revealed that the initial thermal decomposition temperature and the termination temperature of thermal decomposition of CDA−g−PLLA were lower than CDA; thus, the thermal stability of CDA−g−PLLA became worse, which was attributed to breaking of the hydrogen bonds within the CDA molecule after grafting the flexible chain segment of PLLA. Comparing the TGA and DTG curves of CDA−g−PLLA with different grafting temperatures revealed that the graft copolymers that were grafted at 150 °C (CDA−g−PLLA-150) had the lowest initial degradation temperature of 218.7 °C, which may be due to the highest PLLA grafting rate.

### 3.5. DSC Analysis

The second heating DSC curves of CDA, PLLA, and CDA−g−PLLA with different grafting temperatures are given in Figure 7. As shown in the figure, the T_g_ of CDA is 193.17 °C because the chain entanglement generally increased, making it difficult for the CDA molecular chain to move. Thus, the thermal processing performance of CDA was poor. The T_g_ of CDA−g−PLLA with different grafting temperatures was not much different from that of PLLA, which was approximately 54.2 °C, indicating that the introduction of the flexible PLLA molecular chain reduced the rigidity of the CDA molecular chain.

### 3.6. XRD Analysis

To understand the crystallization properties of CDA, PLLA, and CDA−g−PLLA with different grafting temperatures, XRD was used to measure the samples. As shown in Figure 8, five characteristic peaks were found at 2θ = 8.6°, 10.4°, 13.3°, 18.4°, and 22.5° for CDA, indicating that CDA had crystallization ability. By contrast, most of the grafted CDA−g−PLLA only showed a wider diffraction peak near 20°, indicating that the regularity of the original CDA molecular chains was destroyed after grafting PLLA molecular chains. However, the CDA−g−PLLA that was grafted at 150 °C showed a crystalline peak of PLLA at 16.6°, corresponding to the α (110/200) crystal plane of PLLA. On the basis of the obtained results, the reaction temperature of 150 °C is more favorable for the formation of graft copolymers.

From the aforementioned results, grafting PLLA to CDA can decrease the T_g_ of CDA and improve the thermal processing performance, however, the introduction of the flexible PLLA molecular chain reduced the rigidity of the CDA molecular chain, so it will be easier for chains to slip. This may lead to worsening mechanical properties and softening temperature of the prepared graft copolymers.

## 4. Conclusions

CDA−g−PLLA graft copolymers were synthesized using a reactor under nitrogen protection, and the influencing factors of the grafting process were investigated. From the above results, the grafting rate was the highest under the condition with the L-LA/CDA feeding mass ratio of 4:1, reaction time of 90 min, and reaction temperature of 150 °C. The PLLA molecular chains were successfully grafted onto the CDA molecular chain, and no PLLA homopolymer was found in the grafting product after purification. The lowest initial weight loss temperature of CDA−g−PLLA was 218.7 °C, and a certain microscopic phase separation occurred within the molecule. The XRD and DSC curves revealed that the regularity of the original CDA molecular chains was destroyed after grafting PLLA molecular chains, leading to the decrease in the rigidity and thermal stability of the graft copolymers. Thus, the T_g_ of CDA−g−PLLA was 54.2 °C, which was much lower than that of CDA (138.97 °C). In this study, the grafting technique provided a facile and effective method to modify the properties of CDA and it has provided process reference for the synthesis of CDA−g−PDLA and CDA−g−scPLA materials. This will further broaden the application field of CDA in thermal processing.

## Figures and Tables

**Figure 1 polymers-15-00143-f001:**
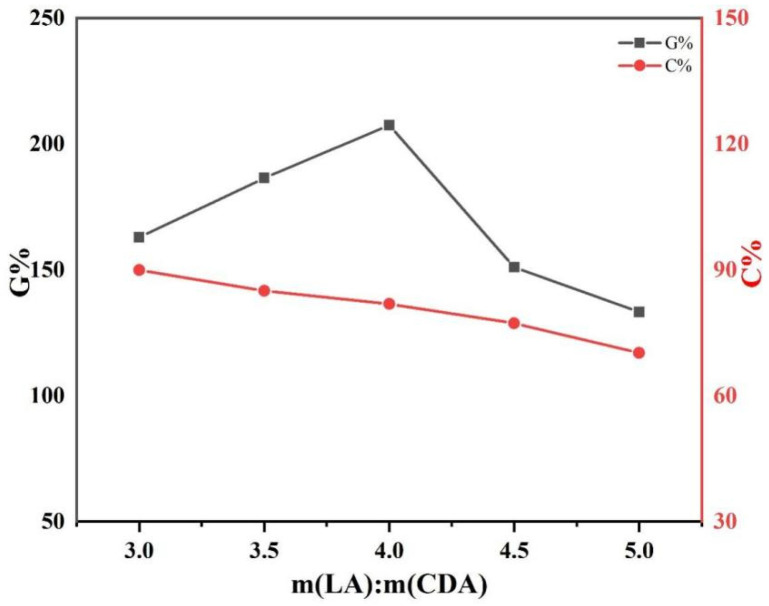
Grafting rate and monomer conversion rate of CDA−g−PLLA under different feeding mass ratio conditions at reaction temperatures of 130 °C for 1 h.

**Figure 2 polymers-15-00143-f002:**
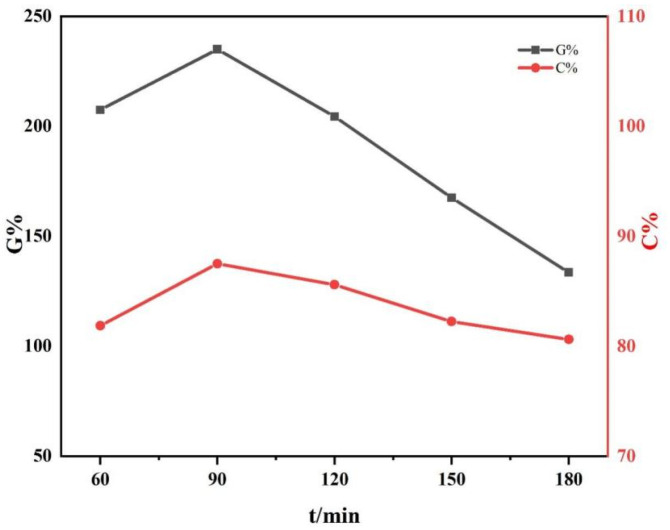
Grafting rate and monomer conversion rate of CDA−g−PLLA at different reaction times by reacting L-LA/CDA at a feeding mass ratio of 4:1 at 130 °C.

**Figure 3 polymers-15-00143-f003:**
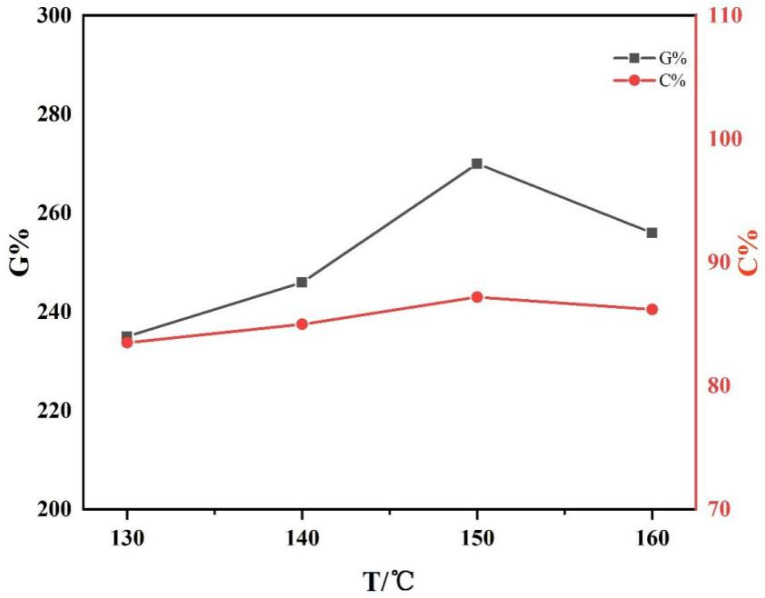
Grafting rate and monomer conversion of CDA−g−PLLA at different reaction temperatures under the reaction of L-LA/CDA feeding mass ratio of 4:1 for 90 min.

**Figure 4 polymers-15-00143-f004:**
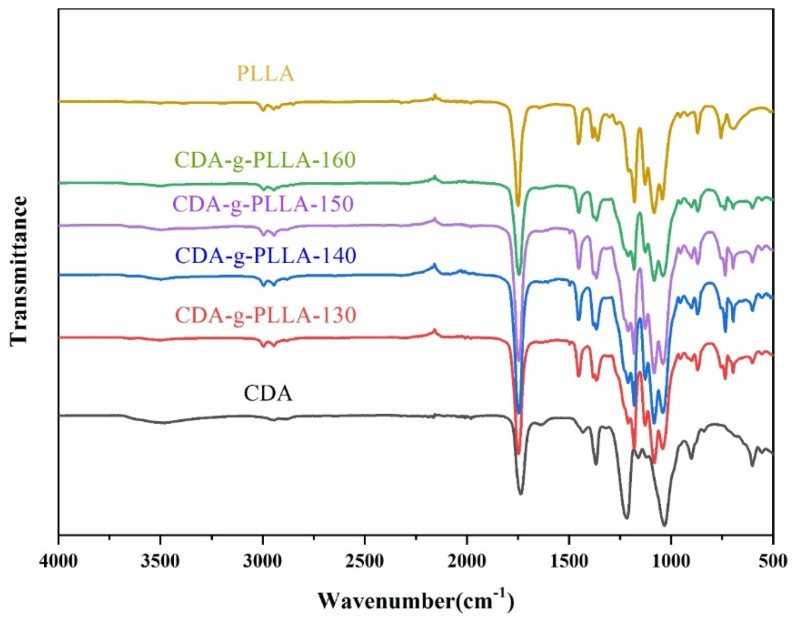
FTIR spectra of CDA, PLLA, and CDA−g−PLLA with different grafting temperatures.

**Figure 5 polymers-15-00143-f005:**
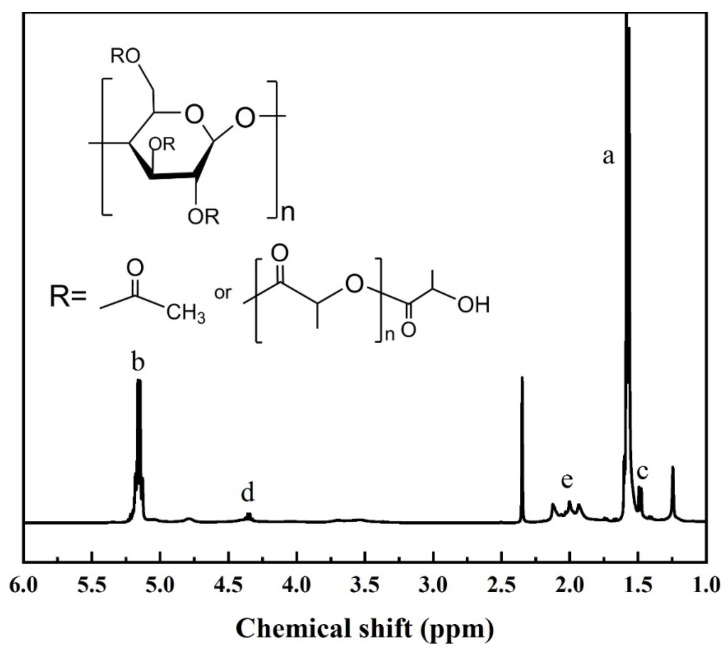
^1^H NMR spectrum and structural formula of CDA−g−PLLA.

**Figure 6 polymers-15-00143-f006:**
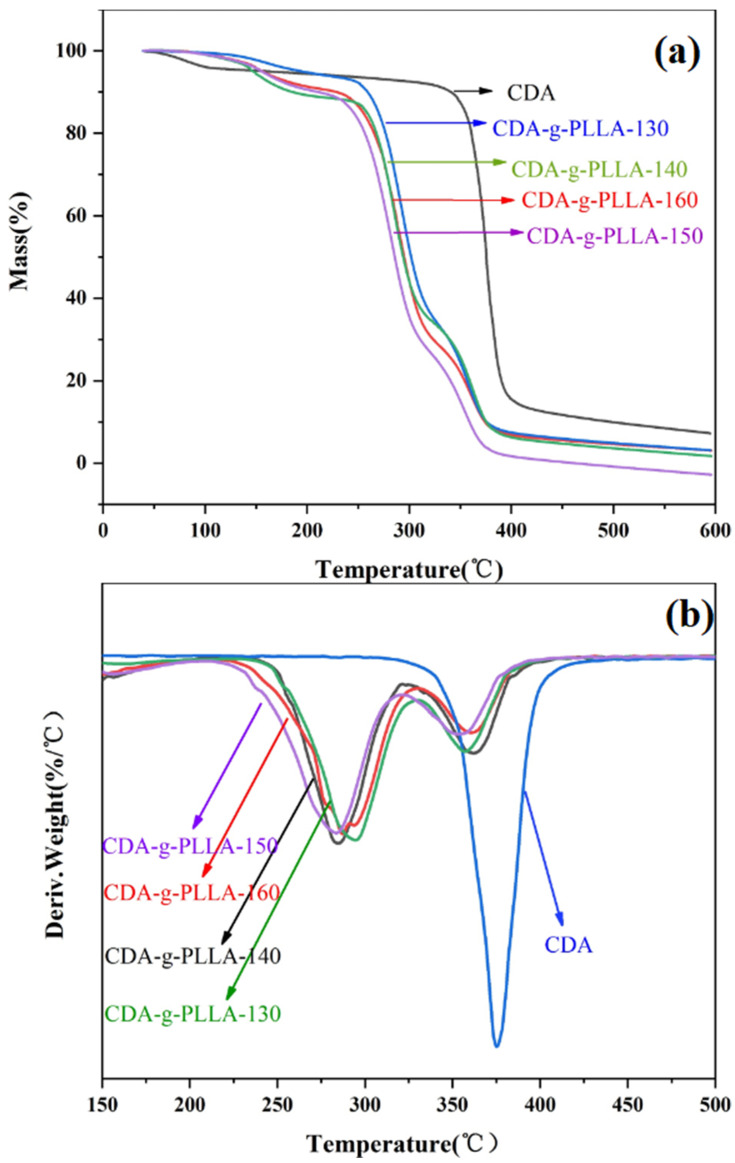
TGA (**a**) and DTG (**b**) curves of CDA and CDA−g−PLLA.

**Figure 7 polymers-15-00143-f007:**
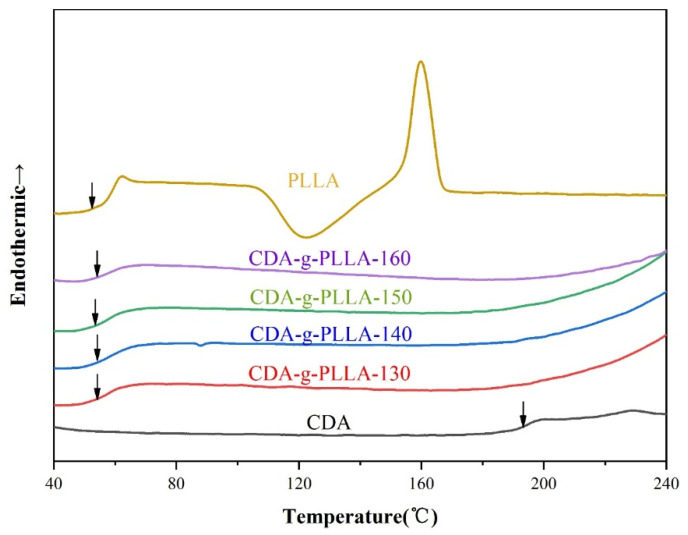
DSC plots of CDA, PLLA, and CDA−g−PLLA with different grafting temperatures.

**Figure 8 polymers-15-00143-f008:**
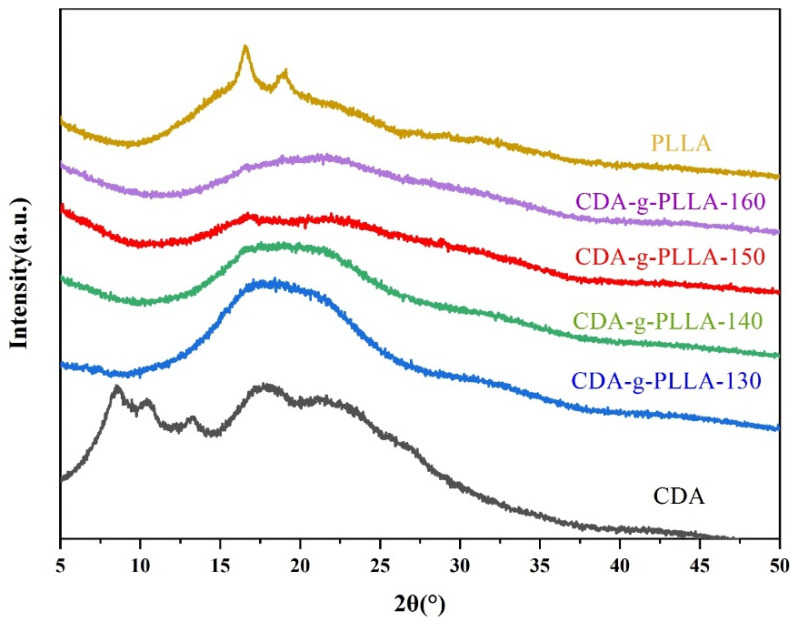
XRD patterns of CDA, PLLA, and CDA−g−PLLA with different grafting temperatures.

**Table 1 polymers-15-00143-t001:** Compositional parameters of CDA−g−PLLA graft copolymers.

Sample	MS ^a^	Lactyl DS ^a^	DPs ^a^
CDA−g−PLLA−130	13.61	0.229	59.42
CDA−g−PLLA−140	10.12	0.234	43.24
CDA−g−PLLA−150	19.38	0.413	47.27
CDA−g−PLLA−160	13.78	0.317	43.46

^a^ Determined by ^1^H NMR.

## Data Availability

Data available on request.

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
