# Peer review of "Synthesis and Characterization of Cellulose Diacetate-Graft-Polylactide via Solvent-Free Melt Ring-Opening Graft Copolymerization"

_polymers, 2022, doi:10.3390/polym15010143_

Round 1

Reviewer 1 Report

This article is devoted to the preparation and study of cellulose diacetate copolymers. The article in terms of volume and subject matter meets the requirements of the journal. An interesting new functional polymer based on polysaccharides is in the trend of modern trends in macromolecular chemistry. Despite the interest and importance of this study, I recommend correcting the following points:

1. To more clearly define the purpose and possibilities of practical application of the data of this study.

2. Abstract can be expanded.

3. The authors claim that they crushed the CDA to a powder. What is the approximate particle size in a given powder.

4. Methanol is a toxic reagent that causes many ailments in humans. Why was methanol chosen?

5. "3.1" The authors consider various factors influencing the synthesis process. This part requires a more detailed description of the results obtained. Comparison with literary sources is also necessary.

6. NMR analysis is also described too modestly. Needs to be expanded.

7. Due to what is the decrease in the thermal stability of the polymer during its chemical modification?

8. XRD analysis can be described more extensively. Consider various chemical modification options, how and why they increase amorphization. It is also desirable to make an estimate of the crystallinity index for the samples.

9. Please cite: 10.1007/s00226-022-01363-4, 10.1021/acsomega.1c02570.

10. Conclusions can be made more concise.

In general, the work is interesting, significant improvements are needed in the description of the results.

Author Response

Dear Editors and Reviewers:

Thank you for your comments concerning our manuscript entitled “Synthesis and Characterization of Cellulose Diacetate-Graft-Polylactide via Solvent-Free Melt Ring-Opening Graft Copolymerization”. (Manuscript ID: polymers-2112232). We have studied comments carefully and have made correction which we hope meet with approval. Revised portion were marked in red in the paper. The main corrections in the paper and the responses to the reviewer’s comments are as following:

Reviewer 1: 

Q1. To more clearly define the purpose and possibilities of practical application of the data of this study.

Response:We are grateful for the suggestion. To be more clearly and in accordance with the reviewer concerns, we have added a more detailed interpretation regarding the purpose and possibilities of practical application of the data of this study. More detailed statistical analysis was added in the manuscript.

Q2. Abstract can be expanded.

Response: Thanks for reviewer’s comment, we have expanded the abstract to 210 words.

Q3. The authors claim that they crushed the CDA to a powder. What is the approximate particle size in a given powder.

Response: We are very sorry for our negligence. The approximate particle size in a given powder is 359.04 μm. We have added it in our manuscript.

Q4. Methanol is a toxic reagent that causes many ailments in humans. Why was methanol chosen?

Response: A dissolution–precipitation method can be used to separate the graft copolymer from the mixture because anhydrous methanol can completely dissolve L-LA in the crude product, but not CDA-g-PLLA. Moreover, compared with other solvents, we found that the precipitation of the graft copolymer in absolute methanol was faster.

Q5. "3.1" The authors consider various factors influencing the synthesis process. This part requires a more detailed description of the results obtained. Comparison with literary sources is also necessary.

Response: We are grateful for the suggestion. We have added a more detailed description of the results obtained in "3.1", and comparison with literary sources is also added.

Q6. NMR analysis is also described too modestly. Needs to be expanded.

Response: Thank you for your comments and suggestions. We have expanded the 1H NMR analysis.

“Some resonance peaks are apparent for the methyl proton of the acetyl group in CDA at 1.8−2.1 ppm (e) [35].” And “ Then, estimates of the molar ……this result also verified the conclusion obtained in 3.1.3.”

Q7. Due to what is the decrease in the thermal stability of the polymer during its chemical modification?

Response: The decrease in the thermal stability of the polymer during its chemical modification is due to the regularity of the original CDA molecular chains being destroyed after grafting PLLA molecular chains. This result can be obtained from the XRD analysis

Q8. XRD analysis can be described more extensively. Consider various chemical modification options, how and why they increase amorphization. It is also desirable to make an estimate of the crystallinity index for the samples.

Response: The PLLA molecular chains were grafted onto the CDA molecular chain, and the regularity of the original CDA molecular chains was destroyed after grafting PLLA molecular chains, so they increased the amorphization. Because there are relatively few molecular chains on the graft, the crystallinity index is too small, and the trend change can be seen from the crystallization peak area.

Q9. Please cite: 10.1007/s00226-022-01363-4, 10.1021/acsomega.1c02570.

Response: Thanks for reviewer’s comment , We have cited this paper (Kazachenko et al, ACS Omega, 2021 ) in our manuscript.

Q10. Conclusions can be made more concise.

Response: Thank you for your suggestion. We have revised the conclusion section.

Reviewer 2 Report

In this submitted manuscript (polymers-2112232), Dr. Pan and coworkers investigated the preparation and characterization of cellulose diacetate (CDA)-grafted-poly(L-lactide) (PLLA). The solvent-free melt ring-opening graft copolymerization was first optimized with various reaction conditions, including the feed ratio of CDA and L-LA, reaction temperature and time. Proton NMR, FTIR, TGA/DTG/DSC, and XRD were applied to characterize the structure and thermal properties of the graft copolymer.

While developing low Tg CDA is challenging and important, the manuscript in its current form has weaknesses and the work falls short in terms of rigor and abundant discussion. Major revisions are recommended before further reconsideration.

Criticisms include:

1. It is recommended to use more rigid descriptions in preparing the manuscript. For example, it has been discussed several times about the feed ratio of CDA and L-LA (lines 20, 77, 97, 113, 146, 149, 152, 158, 168, 172, 185, 190, and 276). What is that ratio? Molar ratio or mass ratio? Even though it was mentioned one time (line 154) in the middle of the manuscript, it is the mass ratio between these two constituents, it still requires to clear state as “mass ratio” throughout the manuscript.

2. Why the L-lactide was used instead of D-lactide in this project?

3. What is the purification operation after preparing the CDA-g-PLLA? There is no description of the purification step in the manuscript, making it hard to reproduce.

4. Grafting PLLA to CDA can decrease the Tg of CDA and improve the thermal processing performance, could this decrease worsen other properties of CDA? Discussions about this possibility should be given in the manuscript.

5. How many scans were used in the FTIR characterization? The authors should provide it in section 2.3.2.

6. The conditions for DTG characterization should be provided in section 2.3.

7. Based on what was discussed, it is not correct to use “CDA” on line 237, but should be “CDA-g-PLLA”.

8. The crystalline peak of CDA-g-PLLA (grafted at 150 °C) at 16.6o is ambiguous from Figure 8. It may not be very sound to claim it as a peak as it is too small.

9. A description of the reactor used in the preparation of grafted copolymer should be provided.

10. The language needs to be improved. There are several grammatical errors in the manuscript. For example, in line 13, it should not be “lower” as there is no comparison in that sentence. The word “low” would be the correct one here. A similar example was also found on line 23. Either changing “lower” to “low” or revising as “has a lower Tg than CDA parent polymer” could make this sentence correct.

 Line 21, the name of PLLA should be poly-L-lactide.

On line 49, the sentence should be revised as “…two and three times, respectively, when the…”.

On line 58, the word “produces” should be revised to “produced” to make it a correct sentence.

On line 86, it is not a complete sentence as what was written in the manuscript “L-LA; it was…”.

It is improper to say “The reaction was stirred and dissolved…” on line 97. How could a reaction be dissolved? The reactants can be dissolved but not the reaction.

What does it mean “2% mass fraction of CDA were added to the reactor” on lines 98-99 and 147-148? Since the CDA has already been added to the reactor, why this sentence was written here? If that is the amount of Sn(Oct)2, then that sentence should be revised, otherwise, it is confusing and not correct.

On line 100, instead of saying “When the reactor was cooling to room temperature”, it should be “When the reactor cooled to room temperature”.

Lines 166-167 and 184-185 could be deleted as the sentences “CDA and L-LA were grafted under nitrogen atmosphere, and Sn(Oct)2 with 2% mass fraction of CDA was used as the catalyst” have been repeated several times.

On line 235, instead of using a semicolon, a period should be used to separate two complete sentences. The same suggestion for line 239.

It is not correct to say “…CDA destroyed the regularity of the original molecular chains after grafting PLLA molecular chains” on lines 266-267. It is recommended to revise as “…the regularity of the original CDA molecular chains was destroyed after grafting PLLA molecular chains”.

On line 276, the sentence “…was the highest at the…” should be revised as “…was the highest under the condition with the…” to make it correct.

Author Response

Dear Editors and Reviewers:

Thank you for your comments concerning our manuscript entitled “Synthesis and Characterization of Cellulose Diacetate-Graft-Polylactide via Solvent-Free Melt Ring-Opening Graft Copolymerization”. (Manuscript ID: polymers-2112232). We have studied comments carefully and have made correction which we hope meet with approval. Revised portion were marked in red in the paper. The main corrections in the paper and the responses to the reviewer’s comments are as following:

Reviewer 2:

Q1. It is recommended to use more rigid descriptions in preparing the manuscript. For example, it has been discussed several times about the feed ratio of CDA and L-LA (lines 20, 77, 97, 113, 146, 149, 152, 158, 168, 172, 185, 190, and 276). What is that ratio? Molar ratio or mass ratio? Even though it was mentioned one time (line 154) in the middle of the manuscript, it is the mass ratio between these two constituents, it still requires to clear state as “mass ratio” throughout the manuscript.

Response: We are grateful for the suggestion. In order to state more clearly, we have changed the “feeding ratio” of the full text to the “feeding mass ratio”.

Q2. Why the L-lactide was used instead of D-lactide in this project?

Response: Because D-lactide is more expensive and their chemical structures are similar, considering the economic factors, we first use L-lactide to explore the grafting process. In the subsequent paper, we will use the best process to prepare CDA-g-PDLA and CDA-g-scPLA.

Q3. What is the purification operation after preparing the CDA-g-PLLA? There is no description of the purification step in the manuscript, making it hard to reproduce.

Response: We have described the purification operation after preparing the CDA-g-PLLA in section 2.2.2.

“The solid reactants were dissolved in trichloromethane and stirred with a magnetic stirrer for 24 hours. After completely dissolving, the solution was slowly poured into anhydrous methanol for precipitation, and the solid products were dried in a vacuum drying oven at 60 °C; solid crude products were finally obtained after filtration. With toluene used as the solvent, the solid crude products were purified in a Soxhlet extractor and refluxed for 24 hours. After drying, the CDA-g-PLLA graft copolymers were obtained”.

Q4. Grafting PLLA to CDA can decrease the Tg of CDA and improve the thermal processing performance, could this decrease worsen other properties of CDA? Discussions about this possibility should be given in the manuscript.

Response: Thank you very much for the excellent and professional revision of our manuscript. We have added some discussions about this possibility in our manuscript.

“From the aforementioned results, grafting PLLA to CDA can decrease the Tg of CDA and improve the thermal processing performance, however, the introduction of the flexible PLLA molecular chain reduced the rigidity of the CDA molecular chain, it will be easier for chains to slip. This may lead to worsening mechanical properties and softening temperature of the graft copolymers.”

Q5. How many scans were used in the FTIR characterization? The authors should provide it in section 2.3.2.

Response: We have added the number of scans in section 2.3.2.

“CDA and purified CDA-g-PLLA were scanned by the ATR method at room temperature using an IS50+IN10 FTIR spectrometer from Nicolet, USA for the samples in the range of 500–4000 cm−1 with an average of 32 scans.”

Q6. The conditions for DTG characterization should be provided in section 2.3.

Response: Because DTG is a derivative of TGA, they used the same test method. We have added DTG to section 2.3.4.

Q7. Based on what was discussed, it is not correct to use “CDA” on line 237, but should be “CDA-g-PLLA”.

Response: Thank you for your correction, we have corrected the expression here.

“Two clear downward trends were observed in the TGA curves of the graft copolymers (corresponding to two peaks in the DTG plots), indicating that the degradation of CDA-g-PLLA was divided into two stages.”

Q8. The crystalline peak of CDA-g-PLLA (grafted at 150 °C) at 16.6o is ambiguous from Figure 8. It may not be very sound to claim it as a peak as it is too small.

Response: Because there are relatively few PLLA molecular chains on the graft, the crystallization peak of PLLA is not obvious, but it is indeed slightly different from the other curves.

Q9. A description of the reactor used in the preparation of grafted copolymer should be provided.

Response: Thanks for reviewer’s comment, we have provided the description of reactor used in the preparation of grafted copolymer in the section “2.2.1”.

Q10. The language needs to be improved. There are several grammatical errors in the manuscript. For example, in line 13, it should not be “lower” as there is no comparison in that sentence. The word “low” would be the correct one here. A similar example was also found on line 23. Either changing “lower” to “low” or revising as “has a lower Tg than CDA parent polymer” could make this sentence correct.

 Line 21, the name of PLLA should be poly-L-lactide.

On line 49, the sentence should be revised as “…two and three times, respectively, when the…”.

On line 58, the word “produces” should be revised to “produced” to make it a correct sentence.

On line 86, it is not a complete sentence as what was written in the manuscript “L-LA; it was…”.

It is improper to say “The reaction was stirred and dissolved…” on line 97. How could a reaction be dissolved? The reactants can be dissolved but not the reaction.

What does it mean “2% mass fraction of CDA were added to the reactor” on lines 98-99 and 147-148? Since the CDA has already been added to the reactor, why this sentence was written here? If that is the amount of Sn(Oct)2, then that sentence should be revised, otherwise, it is confusing and not correct.

On line 100, instead of saying “When the reactor was cooling to room temperature”, it should be “When the reactor cooled to room temperature”.

Lines 166-167 and 184-185 could be deleted as the sentences “CDA and L-LA were grafted under nitrogen atmosphere, and Sn(Oct)2 with 2% mass fraction of CDA was used as the catalyst” have been repeated several times.

On line 235, instead of using a semicolon, a period should be used to separate two complete sentences. The same suggestion for line 239.

It is not correct to say “…CDA destroyed the regularity of the original molecular chains after grafting PLLA molecular chains” on lines 266-267. It is recommended to revise as “…the regularity of the original CDA molecular chains was destroyed after grafting PLLA molecular chains”.

On line 276, the sentence “…was the highest at the…” should be revised as “…was the highest under the condition with the…” to make it correct.

Response: Thank you very much for your valuable comments. We apologize for the language problems in the original manuscript. We have corrected all the grammatical errors in the manuscript mentioned above. And the language presentation was improved with assistance from a native English speaker with appropriate research background.

Round 2

Reviewer 1 Report

accepted

Reviewer 2 Report

In the revised manuscript (polymers-2112232-v2), the authors discussed and provided sufficient explanations for the comments raised by reviewers. The quality of the manuscript has been improved evidently, and I'm happy to recommend accepting it in its present form.